# Practice Guidelines for Monitoring Neuromuscular Blockade—Elements to Change to Increase the Quality of Anesthesiological Procedures and How to Improve the Acceleromyographic Method

**DOI:** 10.3390/jcm13071976

**Published:** 2024-03-29

**Authors:** Urszula Kosciuczuk, Agnieszka Dardzinska, Anna Kasperczuk, Paweł Dzienis, Adam Tomaszuk, Katarzyna Tarnowska, Ewa Rynkiewicz-Szczepanska, Agnieszka Kossakowska, Marta Pryzmont

**Affiliations:** 1Department of Anaesthesiology and Intensive Therapy, Medical University of Bialystok, Kilinskiego Street 1, 15-276 Bialystok, Poland; k.tarnowska79@wp.pl (K.T.); ewaryn@op.pl (E.R.-S.); kossakowska-agnieszka@wp.pl (A.K.); pryzmontmarta@wp.pl (M.P.); 2Faculty of Biocybernetics and Biomedical Engineering, Bialystok University of Technology, 15-276 Bialystok, Poland; a.dardzinska@pb.edu.pl; 3Faculty of Mechanical Engineering, Bialystok University of Technology, 15-351 Bialystok, Poland; a.kasperczuk@pb.edu.pl (A.K.); p.dzienis@pb.edu.pl (P.D.); 4Faculty of Electrical Engineering, Bialystok University of Technology, 15-351 Bialystok, Poland; adam.tomaszuk@pb.edu.pl

**Keywords:** acceleromyography, neuromuscular block, neuromuscular monitoring, train-of-four

## Abstract

Neuromuscular blocking agents are a crucial pharmacological element of general anesthesia. Decades of observations and scientific studies have resulted in the identification of many risks associated with the uncontrolled use of neuromuscular blocking agents during general anesthesia or an incomplete reversal of neuromuscular blockade in the postoperative period. Residual relaxation and acute postoperative respiratory depression are the most serious consequences. Cyclic recommendations have been developed by anesthesiology societies from many European countries as well as from the United States and New Zealand. The newest recommendations from the American Society of Anesthesiologists and the European Society of Anesthesiology were published in 2023. These publications contain very detailed recommendations for monitoring the dosage of skeletal muscle relaxants in the different stages of anesthesia—induction, maintenance and recovery, and the postoperative period. Additionally, there are recommendations for various special situations (for example, rapid sequence induction) and patient populations (for example, those with organ failure, obesity, etc.). The guidelines also refer to pharmacological drugs for reversing the neuromuscular transmission blockade. Despite the development of several editions of recommendations for monitoring neuromuscular blockade, observational and survey data indicate that their practical implementation is very limited. The aim of this review was to present the professional, technical, and technological factors that limit the implementation of these recommendations in order to improve the implementation of the guidelines and increase the quality of anesthesiological procedures and perioperative safety.

## 1. Introduction

Neuromuscular blocking agents were introduced as a pharmacological methodology for anesthesia in 1946 by the scientific activities of T. Cecil Gray and Halton. They developed the Liverpool anesthetic technique, which involves using opioid analgesics (pethidine) and skeletal muscle relaxants (d-tubocurarine) to reduce the doses of inhalational and intravenous anesthetics. These crucial clinical observations are still very relevant today, and Gray and Halton’s principles form the theory of combined anesthesia [1,2,3]. Additional practical uses for skeletal muscle relaxants include enabling atraumatic tracheal intubation and a passive position and the applied conditions during surgery, especially during surgical activities involving body cavities [4,5,6,7,8,9]. Modern surgery would not be possible without the availability of skeletal muscle relaxants, and curare (the first muscle relaxant) is as important in anesthesiology as antiseptic management in surgical procedures [10].

Decades of observations and scientific studies have identified many risks associated with the use of neuromuscular blocking agents during general anesthesia and in the postoperative period. Residual relaxation and acute postoperative respiratory depression are the most serious consequences. In the 1950s, the inadequate recovery of neuromuscular transmission and the depression of the acetylcholine cholinesterase, as well as the use of neuromuscular blocking agents, were associated with perioperative mortality 1:370 anesthetics [11]. Epidemiological studies from the 1980s indicated that postoperative residual neuromuscular blockade occurs in approximately 40–60% of patients. Recent observational studies have found that the current frequency is up to 32%, but the use of clinical (qualitative) and quantitative observations can reduce this frequency to 18 and 27%, respectively [10,11,12]. Cardiac and thoracic surgery, morbidly obese populations (BMI more than 40 kg/m^2^), end-stage renal failure, liver dysfunction, and neuromuscular disorders are factors that contribute to ahigher risk of inadequate reversal neuromuscular block, and pediatric and geriatric patients are also at higher risk [13,14].

Due to the occurrence of complications from the uncontrolled use and inadequate assessment of the degree of muscle relaxation, guidelines have been developed for monitoring the effect of skeletal muscle relaxation and the use of muscle relaxant antagonists. The validity of such a procedure has been confirmed by the recommendations of many societies [15,16,17,18,19,20,21,22,23].

The recommendations of the American Society of Anesthesiologists (ASA) and the European Society of Anaesthesiologists (ESA), published in 2023, deserve special attention [1,2]. They indicate the necessity of monitoring the degree of skeletal muscle relaxation through objective methods and train-of-four (TOF) stimulation. It is necessary to monitor the level of skeletal muscle tone when administering relaxants during general anesthesia in an operating theater or a postoperative care unit. The authors point out that monitoring with a peripheral nerve stimulator is a more accurate method than clinical assessments. The gold standard is the evaluation of the function of the thumb adductor muscle in response to stimulation with the TOF pattern—the stimulation of the ulnar nerve with a sequence of four pulses at a frequency of 2 Hz, spaced 0.5 s apart with 15 s intervals [1,2]. Cyclic recommendations have also been developed by anesthesiology societies from the United States, New Zealand, and European countries [2,15,16,17,18,19,20]. The current legislation in Poland requires that the anesthesia workstation be equipped with devices for monitoring the degree of relaxation. The details of the procedure were determined based on the recommendations of the Society of Anesthesiology and Intensive Therapy [24,25,26,27].

The latest ASA and ESA guidelines concern the monitoring of the effects of skeletal muscle relaxants during anesthesia in the operating theater and Post Anesthesia Care Unit (PACU). Additionally, they also provide recommendations for reversing the neuromuscular transmission blockade and methods of protection for residual paralysis.The summary of the ASA recommendations indicates that the use of quantitative assessments of skeletal muscle (e.g., m. adductor pollicis) tone when using neuromuscular blocking agents has greater clinical value than assessments based on clinical symptoms, and this approach can avoid the occurrence of residual paralysis. There are also strong recommendations for extubation when a TOF ratio ≥ 0.9 is achieved. Recommendations for reversing neuromuscular block are also presented: sugammadex is recommended over neostigmine in deep, moderate, and shallow degrees of neuromuscular block induced by rocuronium. There are no strong recommendations for the use of neostigmine as an alternative to sugammadex in minimal neuromuscular block. Additionally, when using atracurium and cis atracurium, a reversal of the neuromuscular blockade is recommended for a minimal block, and if neuromuscular monitoring is not available, extubation is recommended 10 min after the administration of the acetylcholinesterase inhibitor. If muscle relaxation monitoring is available, extubation is possible if the TOF ratio is ≥0.9 [1].

The ESA guidelines present detailed recommendations for the use of skeletal muscle relaxants (for intubation, reducing larynx/tracheal trauma, and rapid sequence induction) and for reversing neuromuscular blockade. According to these recommendations, sugammadex should be used to antagonize neuromuscular blockade induced by amino steroid relaxants in deep, moderate, and shallow block. There should be a spontaneous return of neuromuscular transmission to a TOF ratio > 0.2, which is considered the borderline level for the use of neostigmine. In particular, the ESA guidelines refer to the prevention of residual paralysis and highlight three important recommendations: stimulation of the ulnar nerve should be used to quantify the adductor pollicis tone; the use of sugammadex is possible with deep, moderate, and shallow neuromuscular blockade; and neostigmine should be used to achieve a TOF ratio of 0.2 and monitoring should be continued until the neuromuscular transmission TOF ratio is greater than or equal to 0.9 [2].

The aim of this review was to present the actual clinical situation in the implementation of these recommendations. Secondly, we aimed to identify and describe the professional, technical, and technological factors that are limiting the introduction of these recommendations into practice. We hope that this information will improve the implementation of the guidelines and increase the quality of anesthesiological procedures and perioperative safety.

## 2. Professional Perspectives

Naguib et al. indicated that 9% of American and 19% of European anesthesiologists did not monitor the muscle relaxant used. Observational studies indicate that clinical subjective assessments of the degree of relaxation are more common in the United States, while objective methods are more common in European countries, and the use of skeletal muscle relaxation monitoring is much higher, reaching more than 70% [28]. Other publications have found that 75% of Danish anesthesiologists reported technical problems when using these devices, accounting for 25% of anesthesia equipment [29]. Batistaki et al. reported that 60% of respondents stated that the devices for monitoring the degree of skeletal muscle relaxation should be minimal in the anesthesia workstation [30]. Cyclically implemented recommendations in Great Britain have resulted in a significant increase in the frequency of monitoring the degree of relaxation, from 9 to 31%, and a reduction in the number of respondents who did not routinely use this method, from 62 to 8.9% [13,31,32,33,34].

When evaluating the clinical aspects of the use of devices for monitoring the degree of skeletal muscle relaxation, it was pointed out that there are significant limitations for monitoring short periods of anesthesia that require low doses of muscle relaxants. The lack of monitoring of the degree of skeletal muscle relaxation during general anesthesia has also been attributed to professional experience, knowledge of the pharmacology of skeletal muscle relaxants, and the prevalence of subjective methods of assessing skeletal muscle tone, which have been described as overly complicated. Another important aspect is a lack of training and knowledge of how to use the devices and in recognizing the most common errors. The results of a questionnaire study indicated that 75% of applications had reported technical errors, including variability in readings, inconsistency with clinical observations, and erroneous readings. The importance of cultural factors and standards was also pointed out. Another limitation for the use of muscle relaxation monitoring stemmed from the belief that this method is only for high-risk patients and long-duration surgical procedures. Moreover, the authors emphasized that in many situations, the failure to use skeletal muscle relaxation monitoring devices during general anesthesia was due to ignorance of the legal implications [35,36,37,38,39].

Survey studies in the anesthetic community have also shown that habits and individual experiences with subjective assessments are very strong factors in not monitoring neuromuscular transmission and reversing neuromuscular blockade. In order to improve perioperative anesthetic safety, it is necessary to increase awareness of the risks associated with the uncontrolled use of skeletal muscle relaxants, and to improve the training for the use of medical devices, the forms of self-education, and access to educational materials [26,27,28,29].

## 3. Technical Aspects

In 1958, the use of a peripheral nerve stimulator was suggested to assess muscle tone and diagnose apnea associated with prolonged muscle relaxants. In 1965, Churchill-Davidson indicated that a satisfactory method for assessing neuromuscular block was to perform motor nerve stimulation with an electric current and observe the contractions of the muscles innervated by this nerve [16,17,18,19]. A peripheral nerve stimulator was used to stimulate the ulnar nerve, and the motor responses in the thumb adductor muscle (in the form of thumb movements) were subjectively (visually) observed. In order to use this method as an objective measure of muscle tension, the stimulation process was combined with a visualization and monitoring system [25,26,27].In 1970, Hassan Ali described the first new device for monitoring thumb movement as an element of muscular tone measurements [40]. The first devices for monitoring neuromuscular transmission blockade were separate peripheral nerve stimulators (TOF-Watch, STIMPOD NMS 450, TOF Scan, TOF-Cuf), which later appeared as compatible parts for anesthesia machines—the so-called original equipment manufacturer (OEM) [41,42,43,44].

In a publication by Brull et al., the characteristics of an ideal peripheral nerve stimulator and an ideal monitoring procedure were presented. The technical aspects were as follows: guidelines for proper electrode placement and cutaneous tissue resistance determinations, DC power supply, rectangular-wave pulse, stimulation current in the parameter range of 20–70 mA for a duration of 0.2–0.3 ms, an electrical charge within a range of 4–21 uC with available stimulation patterns for different degrees of post-tetanic count (PTC), tetanic stimulation (TET; 50 Hz), stimulation with a sequence of four pulses at 15 s intervals (TOF), and stimulation with a single 10.1 Hz stimulus (ST). In addition, the authors also noted the importance of the device’s compatibility with the anesthesia machine, the visualization of the applied current intensity, and the presence of an audio system to provide information about the stimulation being performed or errors in the stimulation process. The ideal monitoring of the degree of relaxation of skeletal muscles should additionally allow for calibration to determine the threshold of the supramaximal stimulation for a single stimulation pattern and include elements of electronic memory of the measurements made, with the possibility of reproducing the measurements [5,37].

Various methods are used to monitor neuromuscular transmission. Electromyography (EMG) is based on the assessment of the electrical activity of a contracting muscle under the influence of external stimulation. EMG monitors many different muscles, it does not require freely moving limbs, it is easy and fast set up, and it has a short calibration time.The main disadvantage is possible interference from surgical devices (electrocautery). Moreover, preload is needed and it requires a fixed arm position to perform baseline calibration [14]. The key limitation of EMG is signal drift due to the hand temperature (2–8% decrease in T1 per C increase). Mechanomyography (MMG) assesses the range of isometric muscle contractions (strength of contractions) in response to nerve stimulation. MMG presents muscular tone directly, is not affected by changes in muscular baseline contractivity or immobilization, and does not require preload. The disadvantage isthe cumbersome and time-consuming setup. The most commonclinical application is acceleromyography (AMG), in which the tension of the skeletal muscles describes the acceleration of the contracting muscle, in accordance with Newton’s second law (force is proportional to the mass and its acceleration). AMG devices use a piezoelectric transducer attached to the thumb. Muscle movements generate an electrical impulse in the piezoelectric element, which can be presented. Comparisons of the application of the various methods for monitoring neuromuscular transmission indicate that the accelerometric method has the best technical specifications: simple elements, measurement location, unaffected by external factors (e.g., body temperature, additional substances increasing conductivity), and suitability for various muscle groups. AMG requires the immobilization of the upper limb to record signals from the m. adductor pollicis, and preload is needed to reduce the measurement variability. The most frequent difficulties experienced during AMG included terror massages, fluctuating TOF values, hand movements, and calibration problems [14]. It is extremely important to note that, the results of EMG, MMG, and AMG are not equivalent [17,18,19,20,21].

Thomsen et al. pointed out the importance of technical factors that affect the clinical usability of the obtained results. The most frequently reported problems and limitations of the accelerometric method are the inability to move the device during stimulation, interference from the device’s motion sensor on the readings, the lack of indication of when the device will perform stimulation or when the next stimulation is expected, and additionally, with stimulations requiring calibration, the device being able to indicate a primary value above 100% [33]. Fuchs-Bruder et al. presented the technical aspects and advantages and disadvantages of the implementation of AMG. A TOF ratio above 1.0 raises significant doubts about the measurements and their clinical value. The recommendations indicate the use of a normalized TOF ratio, which is a mathematical calculation of recovery TOF/control TOF. The authors also drew attention to technological pitfalls, as the available solutions when selecting and assessing T4/T2 when the T2 amplitude is greater than T1 or T4/T1 do not give TOF ratio values higher than 1.0 [15,16,17,18,40,45].

Further technical limitations are the perception of an inconsistency in the results after correct stimulation with the recorded motor response, the instability of the hand placement, the interaction among multiple sensors on the patient’s hand, and the lack of error signaling [30,41,42].

## 4. Physical Aspects

Numerous guidelines recommend the accelerometric method with a TOF pattern stimulation of the ulnar nerve and measurements of the motor reaction of the thumb adductor [1,2,3,4]. TOF stimulation is characterized by the use of four equal electrical pulses at a frequency of 2 Hz, spaced 0.5 s apart. The TOF ratio is a numerical value and indicates the ratio of the motor response after the fourth stimulation (T4) with respect to the motor response after the first stimulation (T1) [1,2,3,4].

The physical principle of the TOF test is based on a certain amount of electrical charge in four impulses being applied to the patient’s skin (Figure 1). The electrical impulses excite the nerve, producing a detectable motion of the thumb. The motion is further detected by an accelerometric sensor and analyzed using the associated digital monitoring equipment.

There are several technical aspects of TOF measurements that need to be considered to improve the reliability of the measuring device. The parameters of the TOF impulses need to be adjusted individually by changing the duration of the impulse *T_i_* as well as the value of the excitation current *I_i_* to match the patient’s skin impedance *Z_s_* of several kilo-ohms on the prepared skin [40]. The energy of the electrical impulses has to be limited to a certain magnitude to prevent damage to the patient’s tissues. The energy *E_TOF_* applied to the skin during a single TOF measurement can be calculated as
(1)$$E_{TOF}=4Z_{S}\int_{0}^{T_{i}}{I_{i}}^{2}dt=4Z_{S}{{T_{i}I}_{i}}^{2}$$

Therefore, based on Equation (1), for commonly used ranges of excitation current, durations of impulses, and the average skin impedance of 5 kΩ [46], one can obtain the energy of a single TOF measurement within the range of 0.5–12.2 µJ. The movement of the thumb caused by the electrical impulses applied to the nerve is detected by an accelerometer. The signal from the accelerometric unit needs to be carefully filtered before further analysis. The mathematical principles for determining the TOF are given below. For a three-axis accelerometric unit (Figure 2), the output signal is composed of three readouts from three axes.
(2)$$\vec{a}=\left[\begin{array}{c} a_{x}\\ 0\\ 0 \end{array}\right]+\left[\begin{array}{c} 0\\ a_{y}\\ 0 \end{array}\right]+\left[\begin{array}{c} 0\\ 0\\ a_{z} \end{array}\right]=\left[\begin{array}{c} a_{x}\\ a_{y}\\ a_{z} \end{array}\right]$$

From Equation (2), the total magnitude of acceleration is given by
(3)$$\left|\vec{a}\right|=\sqrt{{a_{x}}^{2}+{a_{y}}^{2}+{a_{z}}^{2}}$$

Figure 3 shows a raw instantaneous accelerometric signal from a three-axis accelerometer applied to the thumb, showing three consecutive thumb movements.

The TOF measurement is the ratio between the acceleration produced by the first impulse and the acceleration produced by the fourth impulse, which is given by
(4)$$TOF=\frac{max(\left|\vec{a_{1}}\right|)}{max(\left|\vec{a_{4}}\right|)}$$

For reliable TOF measurements, it is important to properly position the stimulus electrodes right above the stimulated nerve. Any unintentional displacement or touching of the patient’s hand can affect the measurements.

From a legal perspective, there is a range of standards that should be considered to ensure reliable device operation in a medical environment. Each device distributed within the European Economic Area should possess mandatory CE conformity markings. This implies compliance with certain electrical reliability standards, which are, among others, EN61000-4 [47], concerning electromagnetic compatibility (EMC) and immunity to radio-frequency disturbances, and EN 55011 [48]. Moreover, the software development process for medical equipment needs to be carried out under the risk analysis requirements specified in IEC 62304 [49].

Taking into consideration the points mentioned above, the designers of TOF test equipment face several constraints, such as software reliability and the electromagnetic susceptibility of the device, particularly concerning the operation of the accelerometric sensor, the data analysis and presentation, and storage, as well as the ability to replace only the stimulation electrodes and accelerometric sensor. The implementation of required data-transfer protocols in stand-alone TOF test devices facilitates integration with existing medical equipment. Taking into account human factors during the design process can help in the development of a user-friendly and more reliable device [46,47,48]. Another technical aspect is the total weight of the device itself, including the accelerometric sensor. To achieve measurements with high sensitivity and accuracy, the weight of the device should be as low as possible. As a consequence, very often, the device is mounted on a rubber O-ring. This introduces the possibility of sliding the device over the entire thumb. The best results for muscle relaxation measurements are obtained when the accelerometer is placed on top of the thumb. However, reducing the weight of the device introduces a problem with correctly positioning the device on the finger. When designing a new model, a compromise between its weight and the accelerometer position should be achieved. In addition, new devices should prevent the thumb from being blocked by other fingers [47,48,49,50].

Studies indicate that the following technical aspects are crucial for this method: the location of the stimulating electrodes and adjustment of the stimulation parameters (current, duration of the impulse), the immobilization of the hand (thumb), and the steady position of the accelerometric sensor. These mechanical aspects need to be solved in the future [49,50,51,52].

In the study conducted as part of “The pilot project on the development of cooperation in the field of R&D between business and universities”, which includes the Medical University of Bialystok and Bialystok University of Technology (project number: UDA-RPPD.01.02.01-20-0203/20-00), attempts were made to use innovative construction materials to design splints to stabilize the hand and a method to attach the electrode to the thumb. The proposed solutions are currently being prepared for a patent application. The proposed solutions will be presented in a future paper after obtaining patent protection.

## 5. Conclusions

The use of skeletal muscle relaxants is an inseparable component of general anesthesia. However, anesthetic awareness requires the monitoring of the effects of these pharmacological agents because the consequences of the ineffective reversal of their action, as well as their prolonged effects, are dangerous to the patient’s health and life in the postoperative period. It is very important to implement the recommendations of the American and European Societies of Anesthesiologists.

The presented recommendations have a very important legal significance because they impose an obligation to ensure perioperative safety by monitoring the level of skeletal muscle relaxation. Therefore, it is also important to remember that clinical observations and professional experience are a counterweight to the objective presentation of the anesthetic situation. The creation of subsequent cyclical guidelines has convinced more anesthesiologists follow the recommendations. The best indicator of the need to implement these recommendations is the reduction in the incidence of residual paralysis. Many local and national Societies of Anesthesiology create recommendations based on the ESA and ASA guidelines, which increases their impact.

Many technological approaches are used for monitoring the patient’s condition during anesthesia, and we are able to monitor all elements of general anesthesia—amnesia, analgesia, and muscle relaxation. Devices for monitoring the degree of skeletal muscle relaxation have been available since the 1950s, but they are still not widely used. In order to improve perioperative anesthetic safety, it is necessary to increase awareness of the risks associated with the uncontrolled use of skeletal muscle relaxants, and to improve training for the use of medical devices, the forms of self-education, and access to educational materials. These elements provide a great opportunity to change anesthetic practice. In our opinion, it is also important to develop new technological solutions to minimize erroneous measurements and adapt the devices to the circumstances of use.

In this article, we presented the professional, technical, and technological factors that are limiting the implementation of recommendations. We hope that new technological ideas will be a major step towards increasing the frequency of the degree of relaxation monitoring and changing the practices and professional habits of anesthesiologists. Moreover, the serious need to implement the recommendations for monitoring the degree of skeletal muscle relaxation in anesthetic practice and the multi-faceted promotion of the modification of anesthetic behaviors should be emphasized. The joint recommendations of the European and American Societies of Anesthesiologists are important for improving the quality of anesthesiological procedures and perioperative safety.

## Figures and Tables

**Figure 1 jcm-13-01976-f001:**
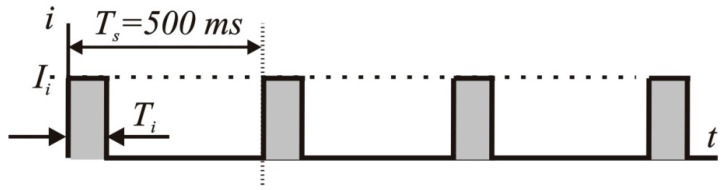
Electrical excitation signal for TOF measurements.

**Figure 2 jcm-13-01976-f002:**
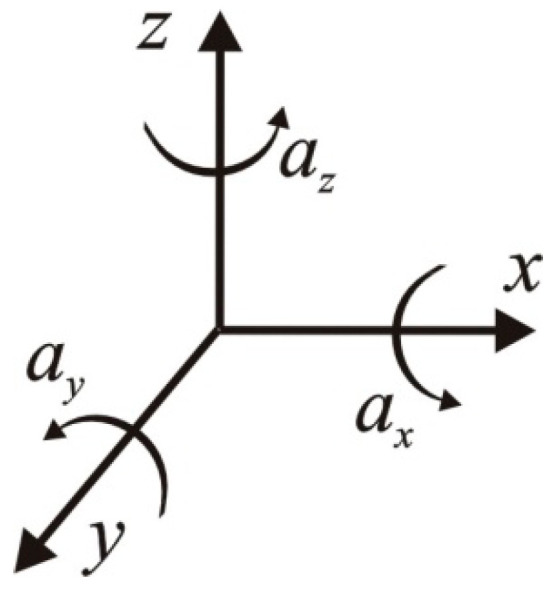
Spatial diagram of the accelerometric signal from a three-axis accelerometer.

**Figure 3 jcm-13-01976-f003:**
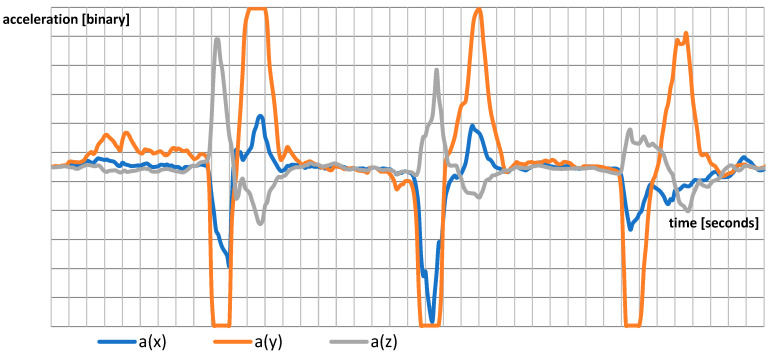
The waveform from three-axis accelerometric unit.

## Data Availability

Data are contained within the article.

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
