# Peer review of "Practice Guidelines for Monitoring Neuromuscular Blockade—Elements to Change to Increase the Quality of Anesthesiological Procedures and How to Improve the Acceleromyographic Method"

_jcm, 2024, doi:10.3390/jcm13071976_

Round 1
Reviewer 1 Report
Comments and Suggestions for Authors
I was pleased to review this paper because the title was very interesting: based on the new international guidelines, what much change to improve neuromuscular block (NMB) management in clinical practice? This is a very important topic.
The authors started with a very good introduction, setting the historical context in which the new ASA and ESAIC guidelines were established.
They exclusively focused on acceleromyography (AMG) -which should be mentioned in the title, intro and conclusion. This is a major limitation of their work because other methods exist (EMG for example) and provide solutions to AMG issues.
The practical perspectives (I would prefer the “professional” perspective) present the problems related to the anesthesiologist malpractice -some of the reasons which limit the guidelines application in clinical practice. I regret that there is no (new) solution: wasn’t the aim of this work to improve the implementation of the new guidelines?
The technical aspects present the historical basis of NMB monitoring, referenced the characteristics of the ideal peripheral nerve monitor, and focused on AMG technical factors affecting the usefulness of results. The baseline reverse fade (TOF ratio > 1.0) is briefly mentioned (line 124). It is one of the main issues related to AMG: I regret that there is no longer consideration, and no solution at all! The authors should refer to normalization and other ways to overcome this problem.
The physical / technological aspects go into TOF and AMG principle of operation. Although interesting, there is few solutions. Particularly, the authors did not refer to the Good Clinical Research Practice (Fuchs-Buder T et al. Good clinical research practice (GCRP) in pharmacodynamic studies of neuromuscular blocking agents III: the 2023 Geneva revision. Acta Anaesthesiol Scand 2023;67:994-1017) that stated how to properly use NMB monitoring.
Another issue of AMG is the installation of the hand during surgery and the problems induced when the arms are fixed alongside the body under the surgical sheets because the thumb cannot move freely (because it is blocked by the other fingers, but not only). Even if this major problem is mentioned in line 163 and 181, it is clearly insufficiently considered, and no solution is proposed. The authors should read about hand adaptor, preload, and TOF-Tube.
This is my major concern with this work: even if the authors focused on AMG, some major problems (limiting its use in clinical practice) are not developed, and no solution is proposed to precisely improve its use -which was the stated aim of this work! In this way, this work finally adds very few knowledges and miss its goal.
Author Response
- Thank you very much for the review. We would like to present changes.
I was pleased to review this paper because the title was very interesting: based on the new international guidelines, what much change to improve neuromuscular block (NMB) management in clinical practice? This is a very important topic.
The authors started with a very good introduction, setting the historical context in which the new ASA and ESAIC guidelines were established.
They exclusively focused on acceleromyography (AMG) -which should be mentioned in the title, intro and conclusion. This is a major limitation of their work because other methods exist (EMG for example) and provide solutions to AMG issues.
We added to Technical aspects: Various methods are used to monitor neuromuscular transmission. Electromyography (EMG) is based on the assessment of the electrical activity of a contracting muscle under the influence of external stimulation. Mechanomyography (MMG), on the other hand, assesses the range of isometric muscle contractions (strength of contractions) in response to nerve stimulation. The most common application is acceleromyography (AMG), in which the tension of the skeletal muscles describes the acceleration of the contracting muscle, in accordance with Newton's second law.
Comparisons of the application of the various methods for monitoring neuromuscular transmission indicate that the accelerometric method has the best technical specifications: simple elements, measurement location, is not affected by external factors (e.g., body temperature, additional substances increasing conductivity), and is suitable for various muscle groups.
The main limitation of MMG is the inability to use alternative muscles, and it is not recommended for scientific research on the effects of muscle relaxants. The key limitation of EMG is signal drift due to the hand temperature (2-8% decrease in T1 per C increase). Like other methods, AMG requires immobilization of the upper limb to record signals from the m. adductor pollicis, and use of preload is needed to reduce measurement variability. It is extremely important that, despite them all being well-known methods, the results of EMG, MMG, and AMG are not equivalent.
The practical perspectives (I would prefer the “professional” perspective) present the problems related to the anesthesiologist malpractice -some of the reasons which limit the guidelines application in clinical practice. I regret that there is no (new) solution: wasn’t the aim of this work to improve the implementation of the new guidelines?
We added to Professional perspective: Survey studies in the anesthetic community have also shown that habits and individual experiences with subjective assessments are very strong factors in not monitoring neuromuscular transmission and reversing neuromuscular blockade. In order to improve perioperative anesthetic safety, it is necessary to increase awareness of the risks associated with the uncontrolled use of skeletal muscle relaxants, and improve the training for the use of medical devices, the forms of self-education, and access to educational materials. Our team strive to develop the new approach, some of the solutions are not to be disclosed at this stage of the project because of ongoing patenting process.
The technical aspects present the historical basis of NMB monitoring, referenced the characteristics of the ideal peripheral nerve monitor, and focused on AMG technical factors affecting the usefulness of results. The baseline reverse fade (TOF ratio > 1.0) is briefly mentioned (line 124). It is one of the main issues related to AMG: I regret that there is no longer consideration, and no solution at all! The authors should refer to normalization and other ways to overcome this problem.
We added to Technical aspects: Fuchs–Bruder et al. presented the technical aspects and advantages and disadvantages of the implementation of AMG. A TOF ratio above 1.0 raises significant doubts about the measurements and their clinical value. The recommendations indicate the use of a normalized TOF ratio, which is a mathematical calculation of recovery TOF/control TOF. The authors also drew attention to technological pitfalls, as the available solutions when selecting and assessing T4/T2 when the T2 amplitude is greater than T1 or T4/T1 do not give TOF ratio values higher than 1.0.
The physical / technological aspects go into TOF and AMG principle of operation. Although interesting, tere is few solutions. Particularly, the authors did not refer to the Good Clinical Research Practice (Fuchs-Buder T et al. Good clinical research practice (GCRP) in pharmacodynamic studies of neuromuscular blocking agents III: the 2023 Geneva revision. Acta Anaesthesiol Scand 2023;67:994-1017) that stated how to properly use NMB monitoring.
All new text about technical aspects are based on Fuchs-Buder T et al. Good clinical research practice (GCRP) in pharmacodynamic studies of neuromuscular blocking agents III: the 2023 Geneva revision. Acta Anaesthesiol Scand 2023;67:994-1017
Another issue of AMG is the installation of the hand during surgery and the problems induced when the arms are fixed alongside the body under the surgical sheets because the thumb cannot move freely (because it is blocked by the other fingers, but not only). Even if this major problem is mentioned in line 163 and 181, it is clearly insufficiently considered, and no solution is proposed. The authors should read about hand adaptor, preload, and TOF-Tube.
Same as above mentioned. The mechanical solution is currently a subject to patent.
This is my major concern with this work: even if the authors focused on AMG, some major problems (limiting its use in clinical practice) are not developed, and no solution is proposed to precisely improve its use -which was the stated aim of this work! In this way, this work finally adds very few knowledges and miss its goal.
The purpose of the paper was mainly to present the issues and outline the problems that currently exist. The solutions that we developed with the team are currently being submitted as a patent application.
We have added a text in the article (lines 187-191):
“In research conducted as part of: “The pilot project on the development of cooperation in the field of R&D between business and universities”, in which take parts: Medical University of Bialystok and Bialystok University of Technology (number of project: UDA-RPPD.01.02.01-20-0203/20-00), attempts were made to use innovative construction materials to design structures of splints for stabilizing the hand and a method of attaching the electrode to the thumb. The proposed solutions are currently being prepared as an application for patent protection. The proposed solutions will be presented in the future paper, after obtaining patent protection.”
Reviewer 2 Report
Comments and Suggestions for Authors
The manuscript is potentially interesting although it report nothing already known in the field.
Major concern
A clinical section is lacking depicting clinical thresholds to allow a safe extubation both in OR and/or in PACU. In addition I would like some comments regarding the use of Sugammadex and other agents reversing paralysis. A comment on the PORC studies is also welcome
Author Response
Thank you very much for the review. We would like to present changes.
A clinical section is lacking depicting clinical thresholds to allow a safe extubation both in OR and/or in PACU. In addition I would like some comments regarding the use of Sugammadex and other agents reversing paralysis. A comment on the PORC studies is also welcome.
We added to Introduction and Professional perspectives: The summary of the ASA recommendations indicates that the use of quantitative assessments of skeletal muscle (e.g., m. adductor pollicis) tone when using neuromuscular blocking agents has greater clinical value than assessments based on clinical symptoms, and this approach can avoid the occurrence of residual paralysis. There are also strong recommendations for extubation when a TOF ratio ≥ 0.9 is achieved. Recommendations for reversing neuromuscular block are also presented: sugammadex is recommended over neostigmine in deep, moderate, and shallow degrees of neuromuscular block induced by rocuronium. There are no strong recommendations for the use of neostigmine as an alternative to sugammadex in minimal neuromuscular block. Additionally, when using atracurium and cis atracurium, a reversal of neuromuscular blockade is recommended for a minimal block, and if neuromuscular monitoring is not available, extubation is recommended 10 minutes after the administration of the acetylcholinesterase inhibitor. If muscle relaxation monitoring is available, extubation is possible if the TOF ratio is ≥ 0.9. In particular, the ESA guidelines refer to the prevention of residual paralysis and highlight three important recommendations: stimulation of the ulnar nerve should be used to quantify the adductor pollicis tone; the use of sugammadex is possible with deep, moderate, and shallow neuromuscular blockade; and neostigmine should be used to achieve a TOF ratio of 0.2 and monitoring should be continued until the neuromuscular transmission TOF ratio is greater than or equal to 0.9.
Reviewer 3 Report
Comments and Suggestions for Authors
- for this type of article (I understand this is some kind of a literature review?), the authors should focus on expanding the introduction section overall
- authors should consider the historical context and acknowledge the contributions of other key figures and developments in the field of anesthesiology
- specify the years or a time period for "current" studies - is it last decade? this decade?
- how clinical and quantitative methods contribute to the reduction of residual blockade?
- summarize key guidelines or recommendations that have significantly influenced practice - a figure or table is needed
- explain the advantages of the accelerometric method with TOF stimulation over other methods
- if authors mention specific national legislation, they should briefly compare it to international standards or practices
- discuss the potential impact of not monitoring muscle relaxant use on patient safety and outcomes
- how cultural attitudes on monitoring and education using these devices can influence their adoption and utilization?
- what are the legal implications of not using muscle relaxation monitoring devices, including potential liability issues?
- how precise TOF measurement can improve patient outcomes and reduce the risk of residual paralysis?
- discuss how how device weight and placement affect the accuracy of muscle relaxation measurements
Title is slightly misleading.
Author Response
Thank you very much for the review. We would like to present changes.
- for this type of article (I understand this is some kind of a literature review?), the authors should focus on expanding the introduction section overall
We added to Introduction:
The aim of this review was to present the actual clinical situation in the implementation of these recommendations. Secondly, we aimed to identify and describe the professional, technical, and technological factors that are limiting the introduction of these recommendations into practice.
- authors should consider the historical context and acknowledge the contributions of other key figures and developments in the field of anesthesiology
We did not decide to present more historical data.
- specify the years or a time period for "current" studies - is it last decade? this decade?
It means this decade. We decided to use the most actual publications in References.
- how clinical and quantitative methods contribute to the reduction of residual blockade?
Part Introduction:
Epidemiological studies from the 1980s indicated that postoperative residual neuromuscular blockade occurs in approximately 40-60% of patients. Recent observational studies have found that the current frequency is up to 32%, but the use of clinical (qualitative) and quantitative observations can reduce this frequency to 18 and 27%, respectively.
Part Professional perspectives:
The summary of the ASA recommendations indicates that the use of quantitative assessments of skeletal muscle (e.g., m. adductor pollicis) tone when using neuromuscular blocking agents has greater clinical value than assessments based on clinical symptoms, and this approach can avoid the occurrence of residual paralysis. There are also strong recommendations for extubation when a TOF ratio ≥ 0.9 is achieved. Recommendations for reversing neuromuscular block are also presented: sugammadex is recommended over neostigmine in deep, moderate, and shallow degrees of neuromuscular block induced by rocuronium. There are no strong recommendations for the use of neostigmine as an alternative to sugammadex in minimal neuromuscular block. Additionally, when using atracurium and cis atracurium, a reversal of neuromuscular blockade is recommended for a minimal block, and if neuromuscular monitoring is not available, extubation is recommended 10 minutes after the administration of the acetylcholinesterase inhibitor. If muscle relaxation monitoring is available, extubation is possible if the TOF ratio is ≥ 0.9. In particular, the ESA guidelines refer to the prevention of residual paralysis and highlight three important recommendations: stimulation of the ulnar nerve should be used to quantify the adductor pollicis tone; the use of sugammadex is possible with deep, moderate, and shallow neuromuscular blockade; and neostigmine should be used to achieve a TOF ratio of 0.2 and monitoring should be continued until the neuromuscular transmission TOF ratio is greater than or equal to 0.9.
- summarize key guidelines or recommendations that have significantly influenced practice - a figure or table is needed
The key ASA and ESA guidelines are described in the Introduction.
- explain the advantages of the accelerometric method with TOF stimulation over other methods
We added to Technical aspects: Various methods are used to monitor neuromuscular transmission. Electromyography (EMG) is based on the assessment of the electrical activity of a contracting muscle under the influence of external stimulation. Mechanomyography (MMG), on the other hand, assesses the range of isometric muscle contractions (strength of contractions) in response to nerve stimulation. The most common application is acceleromyography (AMG), in which the tension of the skeletal muscles describes the acceleration of the contracting muscle, in accordance with Newton's second law.
Comparisons of the application of the various methods for monitoring neuromuscular transmission indicate that the accelerometric method has the best technical specifications: simple elements, measurement location, is not affected by external factors (e.g., body temperature, additional substances increasing conductivity), and is suitable for various muscle groups.
The main limitation of MMG is the inability to use alternative muscles, and it is not recommended for scientific research on the effects of muscle relaxants. The key limitation of EMG is signal drift due to the hand temperature (2-8% decrease in T1 per C increase). Like other methods, AMG requires immobilization of the upper limb to record signals from the m. adductor pollicis, and use of preload is needed to reduce measurement variability. It is extremely important that, despite them all being well-known methods, the results of EMG, MMG, and AMG are not equivalent.
- if authors mention specific national legislation, they should briefly compare it to international standards or practices
We added short explanation in the Introduction: The current legislation in Poland requires that the anesthesia workstation be equipped with devices for monitoring the degree of relaxation [21-24]. The details of the procedure were determined based on the recommendations of the Society of Anesthesiology and Intensive Therapy.
- discuss the potential impact of not monitoring muscle relaxant use on patient safety and outcomes. - how cultural attitudes on monitoring and education using these devices can influence their adoption and utilization,- what are the legal implications of not using muscle relaxation monitoring devices, including potential liability issues?
We added in Professional perspectives:Survey studies in the anesthetic community have also shown that habits and individual experiences with subjective assessments are very strong factors in not monitoring neuromuscular transmission and reversing neuromuscular blockade. In order to improve perioperative anesthetic safety, it is necessary to increase awareness of the risks associated with the uncontrolled use of skeletal muscle relaxants, and improve the training for the use of medical devices, the forms of self-education, and access to educational materials.
- how precise TOF measurement can improve patient outcomes and reduce the risk of residual paralysis?
We presented the ASA and ESA guidelines .
- discuss how device weight and placement affect the accuracy of muscle relaxation measurements
Our mechanical approach allow minimizing the possibility of dispalcement and obstruction.
The displacement of accelerometric sensor (which was already discussed in the paper) along with its mechanical obstruction degrade the TOF measurement, can lead to introduce the errors in determining TOF ratio. First and foremost, proper application of the measuring unit fixture to the patient’s armwith no blocking to the thumb movement should be maintained by the surgical ward staff.
Round 2
Reviewer 1 Report
Comments and Suggestions for Authors
Thank you for this upgraded version of your manuscript.
The title and the conclusions are better related to the content.
Keywords: add acceleromyography
OK for all the new text, but it must be supported by proper references! Especially sentences in lines 74-84, 87-95, 121-122, 155-156 (MMG is not recommended for scientific research ???), 157-158, 158-159, Thomsen 160, Fuchs-Buder 164,...
The references should be revised accordingly.
Comments on the Quality of English Language
Line 98: we hope this info WILL improve...
Discreet inaccuracies might remain in the text, but I'm not qualify to go into langage edition details.
Author Response
Dear Reviewer, Dear Editor,
We would like to thank you for the review.
We have made a analysis of the manuscript and references. We have added more and checked the validity of the citations. We added to keywords - acceleromyography.
Reviewer 2 Report
Comments and Suggestions for Authors
The authors tried to improve the manuscript in a too hasty way and they did not create an exhaustive clinical paragraph as advised. .References of the new added material are also lacking
Author Response
Dear Reviewer, Dear Editor,
We would like to thank you for the review.
We have re-evaluated the manuscript based on the presented review.
However, we have not set a separate paragraph for clinical use. We would like to point out that clinical issues are indicated in the Introduction section - a summary of guidelines for monitoring the dosage of relaxants and preventing residua relaxation. In the Professional perspectives section, we have pointed out erroneous anesthesia practices. We have outlined the advantages and disadvantages of different methods for monitoring skeletal muscle tone in the Technicals aspects section.
In addition, we have analyzed the citations, we have added more citations for topic updates.
Reviewer 3 Report
Comments and Suggestions for Authors
Authors have now prepared the article according to recommendations.
Author Response
Dear Reviewer, Dear Editor,
We would like to thank you for the review.
Round 3
Reviewer 2 Report
Comments and Suggestions for Authors
none